# Coastal flood impacts and lost ecosystem services along Europe's outermost regions and overseas countries and territories

Michalis I. Vousdoukas [1] ✉, Dominik Paprotny[2,3], Lorenzo Mentaschi [4,5], Isavela N. Monioudi [1] & Luc Feyen [6] ✉

Climate change is expected to result in rising seas, exacerbating coastal floods and erosion. Remote islands are projected to be among the most challenged regions, due to their geographic isolation and fragile economies. While, Small Island Developing States have been attracting the attention of scientists and policy makers, Europe's Outermost Regions (ORs) and Overseas Countries and Territories (OCTs) remain poorly studied in terms of their impacts from Sea Level Rise (SLR). Here we carry out a data-modelling framework to comprehensively study risks of flooding, the submergence of flat regions, and coastal erosion along coastlines of ORs and OCTs. Our study shows that under a high emissions scenario by 2150 annually nearly 3000 km² is expected to be flooded, one third of which by tidal flooding, while 150 km² of land will be lost by coastal erosion. This translates into an annual exposure to coastal inundation of up to half a million of people and an economic damage of 5.9 € billion per year - a 40-fold increase from today. Our study shows the increasing benefits in time of stringent climate mitigation, which could nearly halve these impacts in the long run. However, sea levels will continue to rise long after net zero carbon is reached, and so will the consequent impacts, highlighting the critical importance of proactive efforts to increase the resilience of these vulnerable regions against rising seas.

Climate change and consequent sea level rise (SLR) are projected to have a substantial impact on coastal communities through erosion[1,2] and flooding[3-5]. Sea levels have been rising at an accelerating pace[6], and even under stringent climate mitigation policies, the world is committed to a further increase in mean sea level (MSL) of at least 50 cm by the year 2150[7,8]. While damages from coastal flooding are projected to rise sharply worldwide if current coastal protection is not improved, island states are projected to bear the highest burden of climate change[9,10]. Given the small size of their economies and their geographic distance from continental areas, island states can face a wide range of limitations in terms of risk reduction and climate change adaptation[9,11]. Such areas often lack the economic resources to implement all the sustainable solutions needed[12], while general awareness about the challenges they face is limited due to their remoteness, as well as their small size and population, among others. As a result, despite having contributed the least to global greenhouse gas emissions and pollution, they are the most affected areas by climate change[13,14].

[1]Department of Marine Sciences, University of the Aegean, Mytilene, Greece. [2]Institute of Marine and Environmental Sciences, University of Szczecin, Szczecin, Poland. [3]Research Department Transformation Pathways, Potsdam Institute for Climate Impact Research (PIK), Member of the Leibniz Association, Potsdam, Germany. [4]Department of Physics and Astronomy "Augusto Righi" (DIFA), University of Bologna, Bologna, Italy. [5]CMCC Foundation—Euro-Mediterranean Center on Climate Change, Lecce, Italy. [6]European Commission, Joint Research Centre (JRC), Ispra, Italy. ✉e-mail: vousdoukas@gmail.com; luc.feyen@ec.europa.eu

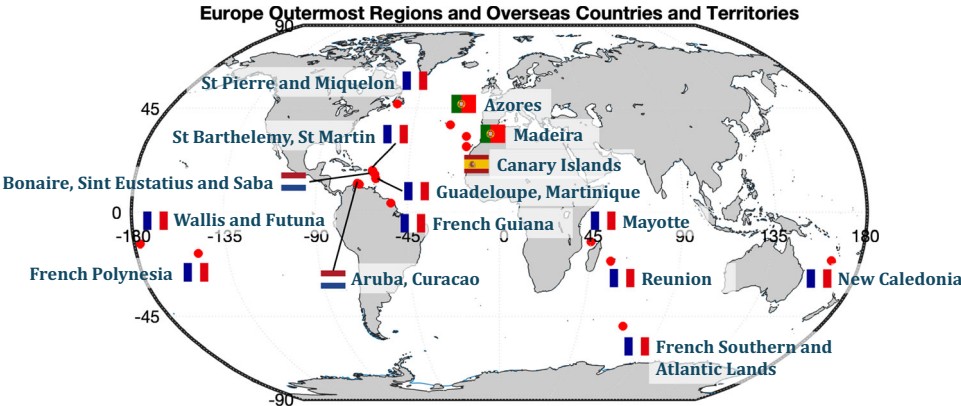

**Fig. 1 | Distribution of Europe's ORs and outstanding territories.** Global map showing the locations of Europe's ORs and Outstanding Territories, with flags denoting the countries they belong to. Base Map was created using MATLAB and the native coastline dataset. The source for the flag icons is: https://github.com/lipis/flag-icons?tab=readme-ov-file. Copyright (c) 2013 Panayiotis Lipiridis.

At the same time, they are among the least studied regions in terms of such challenges worldwide[15].

The above apply also to Europe's 9 outermost regions (ORs) and 13 overseas countries and territories (OCTs), which are scattered across the Atlantic, Caribbean, Indian Ocean, and Pacific (Fig. 1). ORs are an integral part of the EU while OCTs are part of the territory of an EU member state (Denmark, France and the Netherlands), but not of the EU, although they are associated with it. ORs and OCTs are all islands or archipelagos, except for French Guiana, which is located on the north Atlantic coast of South America. They are diverse in geography and geology, ranging from volcanic landscapes to coral reefs, and rich in biodiversity and natural resources. Several ORs and OCTs are low-lying atolls, e.g. in French Polynesia and the Dutch OCTs (e.g. Aruba and Bonaire) and hence are inherently susceptible to coastal erosion and inundation. Economically, they vary from tourism-driven economies in the Canary Islands and French Polynesia to more traditional fishing and agriculture in others (e.g. Saint Pierre and Miquelon and French Guiana). Geographically, they are dispersed, creating challenges and opportunities in connectivity, climate adaptation, and sustainable development.

Despite these differences, their economies rely, to varying degrees, on tourism, agriculture, and fisheries, all sectors profoundly impacted by rising sea levels. Meanwhile, even seemingly elevated regions like the Canaries are not immune, with coastal infrastructure, settlements, beaches, and agricultural land all under increasing pressure. In the present work, we quantify the socio-economic impacts of rising seas for Europe's OCTs and ORs. To that end, we use state-of-the-art estimates of extreme sea levels (ESLs), produced by a coupled wave-ocean circulation model, and the latest SLR projections from IPCC AR6. We apply a hydrodynamic model to simulate coastal flooding, considering the latest available information on the islands' topography, and then combine simulated flooded areas (FAs) with exposure and vulnerability data to estimate flood economic impacts and population exposure. Finally, we assess coastal erosion and the consequent ecosystem service losses in monetary terms.

## Results

### Direct coastal flood impacts

At present, annually 582.29 [370.52–900.15] km² (values in brackets express the very likely range defined by the 5th–95th percentiles) of land along coastlines of ORs and OCTs is expected to be flooded (Fig. 2c), corresponding to 0.10 ± [0.07–0.16%]% of the region's total area. Almost 84% of this total expected annual flooded area (EAFA) is located in OCTs, and 16% in ORs. These FAs correspond to 14,366 ± [9611–39,444] people (Fig. 2b and Table 1), around 0.24% ± [0.16%–0.67%] of the OCTs and ORs population, who are annually exposed to coastal floods, with 88% of the expected annual population exposed (EAPE) living in ORs. The resulting expected annual damage (EAD) of flooding for the OCTs and ORs is 141.9 ± [98.6–215.0]€ million (Fig. 2a and Table 1), of which 69% occurs in ORs. OCTs host several uninhabited or sparsely populated islands (e.g. the French Southern and Atlantic Lands), so FAs do not always translate into population exposure and damage to infrastructure. On the other hand, certain ORs like the Canary Islands, host urban centres in low lying coastal areas, where extreme events can be more impactful economically (Fig. 3a). However, when expressed relative to the size of the economy, impacts in the OCTs (EAD of 0.15% ± [0.06%–0.30%] of GDP) are larger compared to that in the ORs (0.09% ± [0.08%–0.12%] of GDP).

By the year 2050 overall EAFA for all regions increases by more than 70% compared to now and will reach 1020.00 ± [737.83–1364.38], 1043.17 ± [763.42–1386.11], 1057.85 ± [778.02–1403.60], and 1085.11 ± [799.04–1440.66] km², under SSP1-2.6, SSP2-4.5, SSP3-7.0, and SSP5-8.5, respectively (Fig. 2c). These values correspond to around 0.18% of the total area. FAs keep growing in time and will reach 1399.75 ± [767.38–2230.23], 1575.64 ± [925.34–2472.98], 1771.39 ± [1087.30–2744.34], and 1949.07 ± [1231.70–3019.80] km² by 2100 under SSP1-2.6, SSP2-4.5, SSP3-7.0, and SSP5-8.5, respectively. With 0.25–0.35% of the total land area expected to be flooded, this is 2.5–3.5 times more than today. By 2150, the size of FAs will amount to 1757.58 ± [767.38–3129.40], 2105.84 ± [1009.34–3658.46], 2527.97 ± [1305.60–4294.59], and 2838.58 ± [1517.26–4886.42] km², hence ranging between 0.31% and 0.5%, or 5 times higher than today in the worst-case scenario.

As sea levels rise and FAs grow in size, the number of people exposed is projected to increase around 5 times by the year 2050, 10–17 times by 2100 and 15–28 times by 2150 (Fig. 2b). The resulting 2050 EAPE values are between 0.09 and 0.10 million people, depending on the scenario, and by 2150 they are projected to climb to 0.22 ± [0.04–0.47], 0.29 ± [0.09–0.58], 0.37 ± [0.14–0.69], and 0.42 ± [0.18–0.80] million people, under SSP1-2.6, SSP2-4.5, SSP3-7.0, and SSP5-8.5, respectively (Fig. 2b). These values correspond to around 3.77% ± [0.69%–8.03%], 4.92% ± [1.51%–9.77%], 6.23% ± [2.42%–11.74%], and 7.13% ± [3.03%–13.54%] of the total population by 2150, under SSP1-2.6, SSP2-4.5, SSP3-7.0, and SSP5-8.5, respectively.

Direct economic damages increase not only with growing FAs, but also with higher inundation depths, so the relative rise in projected EAD is even more pronounced compared to EAFE and EAPE. By 2050, EAD increases at least seven times compared to present and will reach 1.18 ± [0.54–2.09], 1.24 ± [0.58–2.15], 1.28 ± [0.61–2.19], and

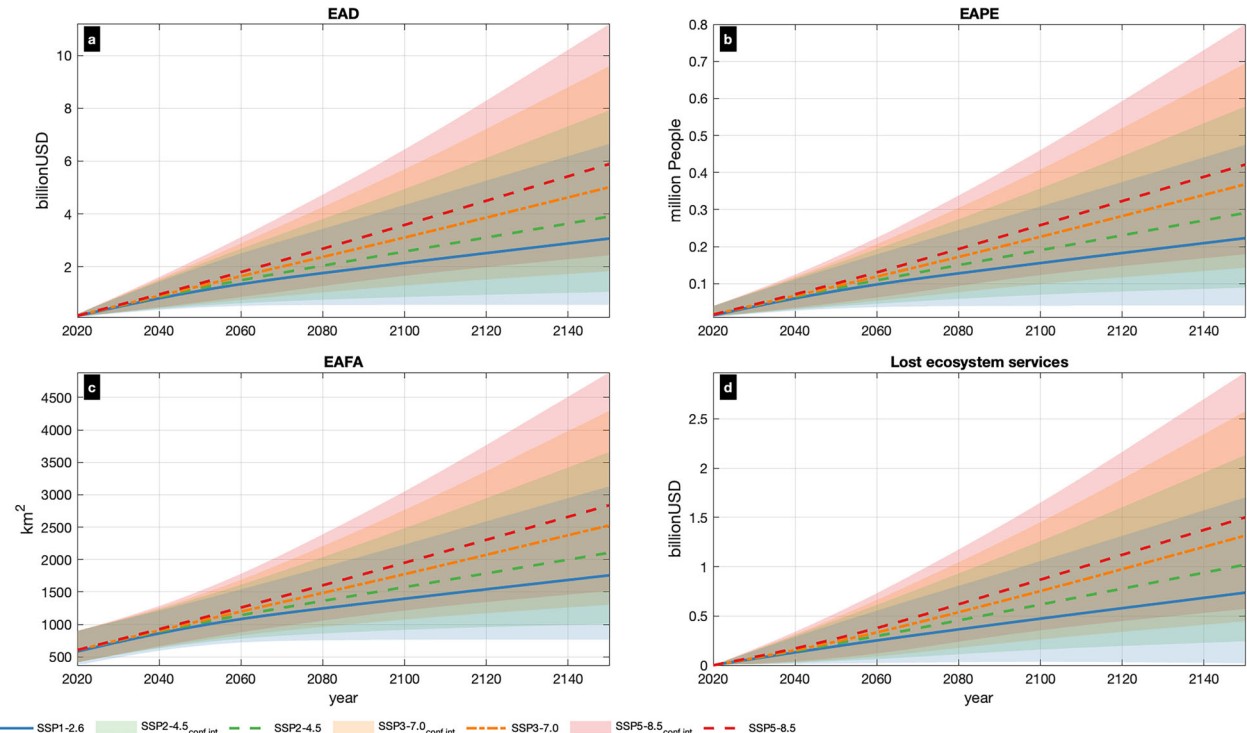

**Fig. 2 | Impacts from coastal floods along Europe's OCTs and ORs will be increasing at least until the year 2150.** Values shown under four scenarios ('low-emissions' (SSP1-2.6), 'moderate-emissions' (SSP2-4.5), 'high-emissions' (SSP3-7.0) and 'very-high-emissions' (SSP5-8.5)). EAD (**a**), EAPE (**b**), EAFA (**c**) and expected annual value of lost ecosystem services (**d**). The lines express the ensemble median projections and the coloured areas the 5th–95th confidence interval.

1.35 ± [0.64–2.28] € billion under SSP1-2.6, SSP2-4.5, SSP3-7.0, and SSP5-8.5, respectively (Fig. 2a). These impacts remain below 1% of GDP, but by the end of the century they will rise 14–24 times compared to present to reach 1.60 ± [0.42–3.23%], 1.92% ± [0.65%–3.66%], 2.31% ± [0.98%–4.20%], and 2.66% ± [1.26%–4.74%] of the GDP under SSP1-2.6, SSP2-4.5, SSP3-7.0, and SSP5-8.5, respectively (Fig. 3d). The corresponding EAD values are 2.15 ± [0.56–4.33], 2.58 ± [0.87–4.92], 3.10 ± [1.31–5.64], and 3.58 ± [1.70–6.36] € billion, respectively. By 2150, projected EAD reaches 3.1–5.9 € billion, corresponding to 2.3–4.4% of the OCT/ORs GDP, or a 21–40 times increase compared to the present-day levels (Fig. 2a).

Overall, OCTs contribute at least 73% of the EAFA by 2050, a number which is reduced to 60.75%–66.10% by the end of the century and to 56.52%–62.12% by 2150. As several of the OCTs are very sparsely populated, ORs contribute to at least 88% of the total EAPE, a number which rises to 93% by the year 2150. The same year, the corresponding median EAPE values are 208,198–395,892; 14,524–25,396 people, for ORs and OCTs, respectively, with a range expressing the scenario uncertainty. Similarly, ORs contribute to most of the 2150 EAD, which is around 2.43–4.77 € billion, compared to 0.64–1.12 € billion for the OCTs, accounting for 2.32%–4.55% and 2.16%–3.79% of the GDP, respectively (Fig. 4c).

Presently, the regions with the highest FA as percentage of the total are Aruba (1.09% ± [0.01%–2.54%]), followed by Bonaire, Sint Eustatius and Saba (1.04% ± [0.53%–5.88%]), French Polynesia (0.94% ± [0.47%–1.39%]), Guadeloupe (0.54% ± [0.35%–1.90%]) and New Caledonia (0.52% ± [0.16%–1.03%]). These values correspond to area equal to 1.97 ± [0.02–4.62], 3.31 ± [1.70–18.76], 33.50 ± [16.89–49.53], 9.04 ± [5.76–31.55], and 96.16 ± [29.47–190.48] km², respectively (Fig. 3c). At least 60% of the FA is found in the French Southern and Antarctic Lands and another 30% is distributed along New Caledonia, French Guiana and French Polynesia.

As for the present-day, also by the end of the century, most of the EAFA is from the French Southern and Antarctic Lands (36.18%–38.87%, depending on the scenario), followed by the French Guiana (24.1%–28.7%) and New Caledonia (17.0%–18.9%). The corresponding median EAFA values are 544.1–705.1, 336.8–559.1, and 265.1–331.2 km², respectively and climb to 651.4–964.3, 484.9–916.9, and 305.1–440.3 km², respectively, by 2150. By the same year, all regions are projected to experience an increase in EAFA between 1 and 40 times compared to the present, depending on the area and scenario. Some smaller islands are projected to have at least 4% of their area annually flooded; e.g. Bonaire, Sint Eustatius and Saba (11.0%–16.6%), Aruba (8.8%–15.5%), Guadeloupe (5.0%–8.6%) and Saint-Martin (4.5%–6.7%). The corresponding median EAFA values are 35.1–53.1, 15.9–28.3, 82.5–143.6, and 2.3–3.4 km², respectively (Fig. 4c). For the above regions, the projected increase in EAFA exceeds 7 and can reach up to 15 times compared to the present. But the highest relative increase in EAFA is projected in Reunion (19.3–40.0 times compared to the present), Bonaire, Sint Eustatius and Saba (9.6–15.0), Martinique (9.4–16.7), Saint-Martin (9.3–14.4), and Guadeloupe (8.1–14.9).

In terms of the present-day EAPE, the region with the highest EAPE as percentage of the total population is Mayotte (0.52% ± [0.38%–0.67%]), followed by Wallis and Futuna (0.52% ± [0.38%–0.65%]), French Guiana (0.45% ± 0.33%–1.10%]), Guadeloupe (0.34% ± [0.22%–1.41%]), and Reunion (0.30% ± [0.23%–0.52%]). Such values correspond to EAPE equal to 1737, 60, 1332, 1327 and 2588 people, respectively. At least one third of the people affected are found in the Canary Islands (4797 people), 18% are in Reunion (2588), and the remaining 31% distributed along Guadeloupe (1327), Mayotte (1737) and French Guiana (1332) (Fig. 4b).

In terms of EAD, about half of the total present-day damages are estimated in New Caledonia and the Canary Islands (24 ± [7.5–50.9] and 40 ± [34.4–46.5] € million, respectively) and one third in French Guiana, Reunion, and French Polynesia (20 ± [16.9–22.9], 17 ± [14.4–19.4], and 15 ± [7.7–23.3] € million, respectively) (Fig. 4a). Some of the above countries feature also among the ones with the

**Table 1 | Area level expected annual values of coastal impacts along Europe's OCTs and ORs by the year 2100 under all four scenarios ('low-emissions' (SSP1-2.6), 'moderate-emissions' (SSP2-4.5), 'high-emissions' (SSP3-7.0), and 'very-high-emissions' (SSP5-8.5))**

| | Country name | SSP1-2.6-2020 | SSP1-2.6-2100 | SSP2-4.5-2100 | SSP3-7.0-2100 | SSP5-8.5-2100 |
|---|---|---|---|---|---|---|
| People affected | Azores | 257 [189–536] | 1899 [582–3621] | 2318 [974–4207] | 2714 [1285–4754] | 3068 [1577–5327] |
| | Canary Islands | 4797 [3475–15,811] | 57,857 [22,881–103,602] | 68,990 [33,303–119,159] | 79,510 [41563–133,697] | 88,908 [49,306–148,916] |
| | French Guiana | 1332 [963–3243] | 13,576 [801–30,286] | 17,643 [4607–35,968] | 21,486 [7625–41,278] | 24,918 [10,453–46,837] |
| | French Southern and Atlantic Lands | 0 [0–0] | 0 [0–0] | 0 [0–0] | 0 [0–0] | 0 [0–0] |
| | Madeira | 248 [179–700] | 2372 [718–4536] | 2899 [1211–5271] | 3396 [1602–5959] | 3841 [1968–6679] |
| | Mayotte | 1417 [1024–1805] | 4951 [898–10,252] | 6241 [2106–12,054] | 7460 [3063–13,739] | 8549 [3960–15,502] |
| | Reunion | 2588 [1940–4465] | 44,946 [2751–100,133] | 58,377 [15,324–118,900] | 71,068 [25,289–136,439] | 82,405 [34,630–154,799] |
| | Saint Pierre and Miquelon | 7 [5–21] | 123 [34–240] | 152 [60–280] | 179 [82–317] | 203 [101–356] |
| | St Barthelemy | 11 [8–17] | 257 [2–747] | 376 [2–913] | 489 [82–1069] | 589 [165–1232] |
| | Wallis and Futuna | 80 [58–100] | 282 [113–504] | 336 [163–580] | 387 [203–650] | 433 [241–724] |
| | Aruba | 286 [3–857] | 3421 [3–6501] | 3799 [1361–6987] | 4575 [2087–8071] | 5308 [2663–9084] |
| | Bonaire, Sint Eustatius and Saba | 71 [36–424] | 853 [36–1432] | 924 [466–1524] | 1070 [602–1728] | 1208 [711–1918] |
| | Curacao | 187 [96–788] | 1332 [96–2133] | 1430 [796–2259] | 1632 [985–2541] | 1823 [1135–2805] |
| | French Polynesia | 339 [173–514] | 755 [173–1116] | 799 [513–1173] | 890 [598–1301] | 976 [666–1420] |
| | Guadeloupe | 1637 [1066–6865] | 15,825 [1066–29,244] | 17,472 [6855–31,359] | 20,852 [10,019–36,081] | 24,045 [12,526–40,495] |
| | Martinique | 429 [177–2184] | 5204 [177–9559] | 5739 [2294–10,245] | 6836 [3320–11,778] | 7872 [4134–13,210] |
| | New Caledonia | 717 [225–2086] | 4649 [225–8002] | 5061 [2409–8530] | 5905 [3199–9710] | 6703 [3825–10,812] |
| | Saint-Martin | 58 [2–318] | 670 [2–1045] | 716 [419–1104] | 810 [508–1236] | 900 [578–1360] |
| | Total | 14,460 [9620–40,733] | 158,974 [30,558–312,953] | 193,273 [72,864–360,515] | 229,260 [102,112–410,349] | 261,748 [128,639–461,477] |
| EAD | Azores | 4.2 [3.5–4.8] | 27.2 [7.8–52.5] | 33.3 [13.5–61.1] | 39.2 [18.1–69.2] | 44.4 [22.4–77.6] |
| | Canary Islands | 40.5 [34.4–46.5] | 352.7 [137.9–633.7] | 421.1 [201.9–729.2] | 485.7 [252.7–818.5] | 543.4 [300.2–912.0] |
| | French Guiana | 19.9 [16.9–22.9] | 239.7 [13.7–535.2] | 311.6 [81.0–635.8] | 379.6 [134.4–729.7] | 440.3 [184.4–828.1] |
| | French Southern and Atlantic Lands | 0.0 [0.0–0.0] | 0.0 [0.0–0.0] | 0.0 [0.0–0.1] | 0.0 [0.0–0.1] | 0.0 [0.0–0.1] |
| | Madeira | 4.5 [3.9–5.2] | 32.5 [9.7–62.3] | 39.8 [16.5–72.5] | 46.6 [21.9–82.0] | 52.7 [26.9–91.9] |
| | Mayotte | 6.9 [5.8–7.9] | 41.3 [0.3–95.0] | 54.4 [12.5–113.2] | 66.7 [22.2–130.3] | 77.8 [31.3–148.2] |
| | Reunion | 16.9 [14.4–19.4] | 430.9 [14.4-1044.4] | 580.2 [101.6-1253.1] | 721.3 [212.3–1448.0] | 847.3 [316.2–1652.2] |
| | Saint Pierre and Miquelon | 0.6 [0.5–0.7] | 7.5 [3.4–12.9] | 8.8 [4.6–14.8] | 10.1 [5.6–16.5] | 11.2 [6.5–18.3] |
| | St Barthelemy | 0.2 [0.2–0.2] | 4.5 [0.2–12.4] | 6.4 [0.3–15.1] | 8.2 [1.7–17.6] | 9.9 [3.1–20.2] |
| | Wallis and Futuna | 0.1 [0.1–0.1] | 0.5 [0.1–1.1] | 0.7 [0.3–1.2] | 0.8 [0.4–1.4] | 0.9 [0.4–1.6] |
| | Aruba | 1.7 [0.0–4.6] | 127.4 [0.0–239.0] | 141.1 [52.9–256.6] | 169.2 [79.2–295.9] | 195.8 [100.0–332.6] |
| | Bonaire, Sint Eustatius and Saba | 1.0 [0.5–6.0] | 49.1 [0.5–85.7] | 53.6 [24.6–91.5] | 62.8 [33.2–104.4] | 71.5 [40.1–116.5] |
| | Curacao | 1.0 [0.5–3.6] | 62.6 [0.5–106.9] | 68.0 [33.0–113.9] | 79.2 [43.4–129.4] | 89.7 [51.7–144.0] |
| | French Polynesia | 15.2 [7.7–23.3] | 35.7 [7.7–54.0] | 37.9 [23.5–56.8] | 42.5 [27.8–63.3] | 46.9 [31.2–69.3] |
| | Guadeloupe | 3.6 [2.3–12.7] | 374.5 [2.3–728.2] | 417.9 [138.1–783.9] | 507.0 [221.5–908.4] | 591.2 [287.6–1024.7] |
| | Martinique | 1.5 [0.6–7.0] | 145.1 [0.6–278.6] | 161.5 [55.9–299.7] | 195.2 [87.4–346.7] | 226.9 [112.3–390.6] |
| | New Caledonia | 47.8 [14.7–99.7] | 461.2 [14.7–812.2] | 504.3 [226.5–867.5] | 592.7 [309.3–991.1] | 676.2 [374.9–1106.5] |
| | Saint-Martin | 0.2 [0.0–1.0] | 21.1 [0.0–34.5] | 22.8 [12.1–36.7] | 26.1 [15.3–41.4] | 29.3 [17.8–45.8] |
| | Total | 165.8 [106.0–265.7] | 2413.6 [213.8–4788.8] | 2863.5 [998.7–5402.7] | 3433.0 [1486.2–6193.8] | 3955.5 [1907.0–6980.0] |

Expected annual number of people exposed and EAD. Values in brackets express the very likely range (5th–95th percentiles).

highest EAD as percentage of the GDP: French Guiana (0.50% ± [0.43%–0.58%]), followed by New Caledonia (0.24% ± [0.07%–0.50%]), Mayotte (0.23% [0.20%–0.27%]), Saint Pierre and Miquelon (0.22% ± [0.19%–0.25%), and French Polynesia (0.20% ± [0.10%–0.30%]) (Fig. 4a).

By the end of the century, most of the EAPE comes from the Canary Islands (34.53%–37.09% of the total, depending on the scenario), followed by the Reunion (28.81%–32.01%). The corresponding median EAPE values are 57,857–88,908 and 44,946–82,405 people, respectively. The same values increase further by 2150 to reach 78,537–138,868 and 69,894–142,677 people, respectively, representing a 15–28 and 26–54 times increase from the present. While the above values express the highest absolute EAPE, the highest EAPE as percentage of the population in 2150 are estimated for Reunion

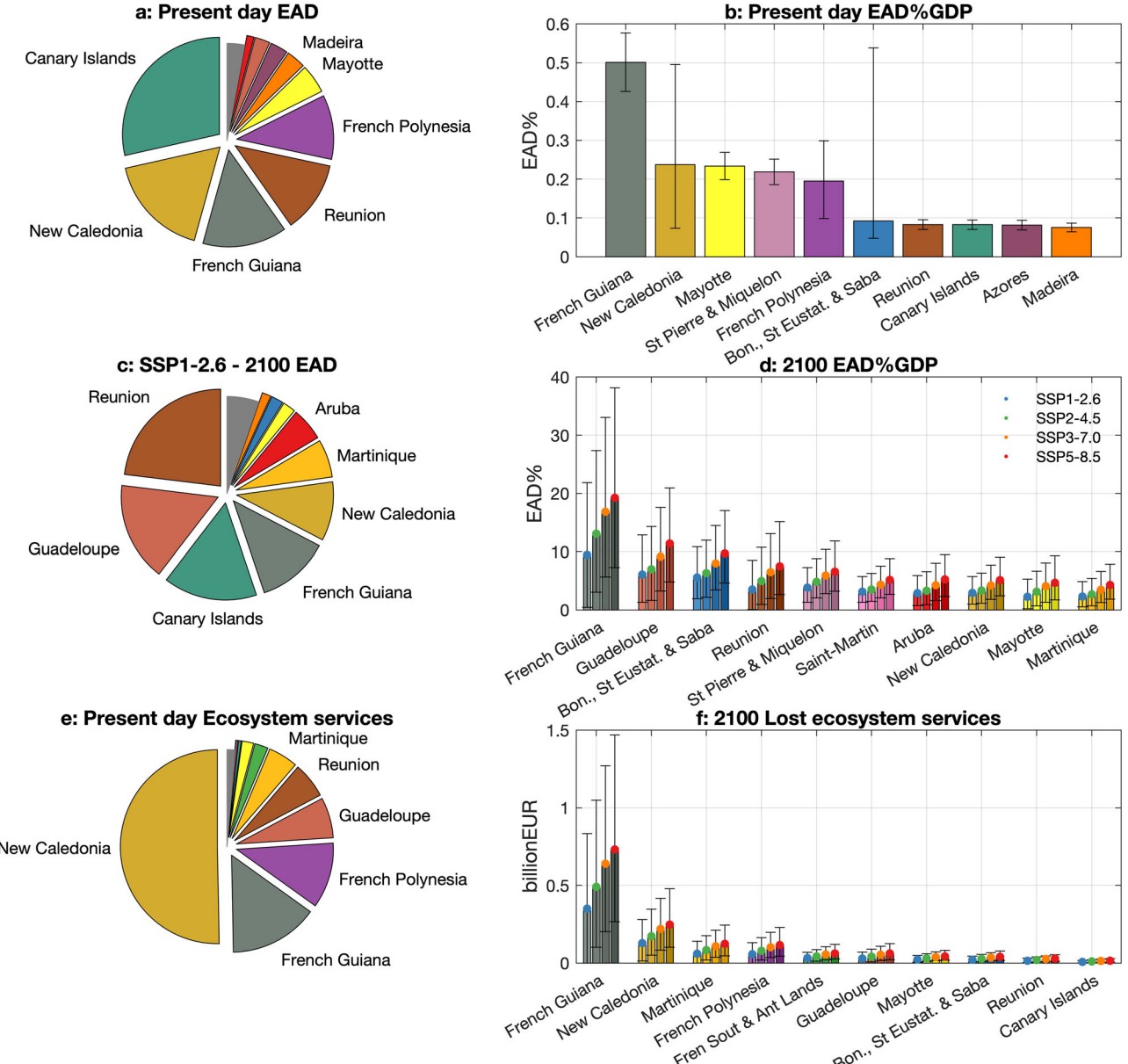

**Fig. 3 | Europe's ORs and OCTs will be increasingly exposed to rising seas during the present century. a** Pie plot indicating the countries with the highest share of the total present-day EAD—colours in (**a**–**d**) are unique for each country and only the countries with the ten highest values are shown. **b** Present-day EAD as percentage of the GDP (bars show median values with black whiskers indicating the 5th–95th confidence interval). **c** Pie plot indicating the highest country contributions to the total 2100 EAD under SSP1-2.6. **d** Ten countries with the highest 2100 EAD as percentage of the GDP (median values in circles with black whiskers indicate the 5th–95th quantile range; bars are grouped in stacks of 5 with one each for the 4 scenarios studied: 'low-emissions' (SSP1-2.6; blue), 'moderate-emissions' (SSP2-4.5; purple), 'high-emissions' (SSP3-7.0; orange), and 'very-high-emissions' (SSP5-8.5; red). **e** Pie plot indicating the highest country contributions to the total present-day Coastal Ecosystem Services. **f** Ten countries with the highest 2100 lost ecosystem services under SSP1-2.6 (median values in circles with black whiskers indicating the 5th–95th quantile range).

(8%–17%), French Guiana (7%–15%), St Barthelemy (5%–12%), Guadeloupe (4%–8%) and Bonaire, Sint Eustatius and Saba (4%–7%), with corresponding median EAPE values being 69,894–142,677; 21,130–43,167; 478–1125; 16,865–30,497; and 1068–1794 people, respectively (Fig. 4b). The highest relative increase in EAPE comes from St Barthelemy (42–100 times from the present), Reunion (26–54), Saint Pierre and Miquelon (25–49), and Canary Islands (15–28).

Guadeloupe accounts for most of the total OCT/ORs EAD by the year 2050 (i.e. around 19%), but after that year it is replaced by Reunion, i.e. 20.00%–23.69% and 23.07%–26.36%, by 2100 and 2150, respectively (range expresses the different emission scenarios). Other countries with high contributions to the total EAD (by 2100 and 2150)

are Guadeloupe (14.9%–16.5%), Canary Islands (14.4%–15.8%), French Guiana (12.2%–13.3%), and New Caledonia (8.6%–9.8%). The corresponding 2150 median EAD values are 0.7–1.5, 0.5–1.0, 0.5–0.9, 0.4–0.8, and 0.3–0.5 € billion, respectively (Fig. 4a). These values correspond to a 41–89, 139–263, 11–20, 18–37, and 11–21 times increase from the present, respectively.

By the end of the century, the highest EAD as percentage is projected in French Guiana (6.0%–11.1%), Guadeloupe (4.5%–7.1%), Bonaire, Sint Eustatius and Saba (4.4%–6.4%), Saint Pierre and Miquelon (2.9%–4.3%), and Saint-Martin (2.5%–3.5%) (e.g. Fig. 3c), corresponding to a 11.1–21.2, 111.3–176.3, 47.6–69.9, 12.1–18.5, and 83.7–116.7 times increase from the present, respectively. Guadeloupe and Saint-Martin

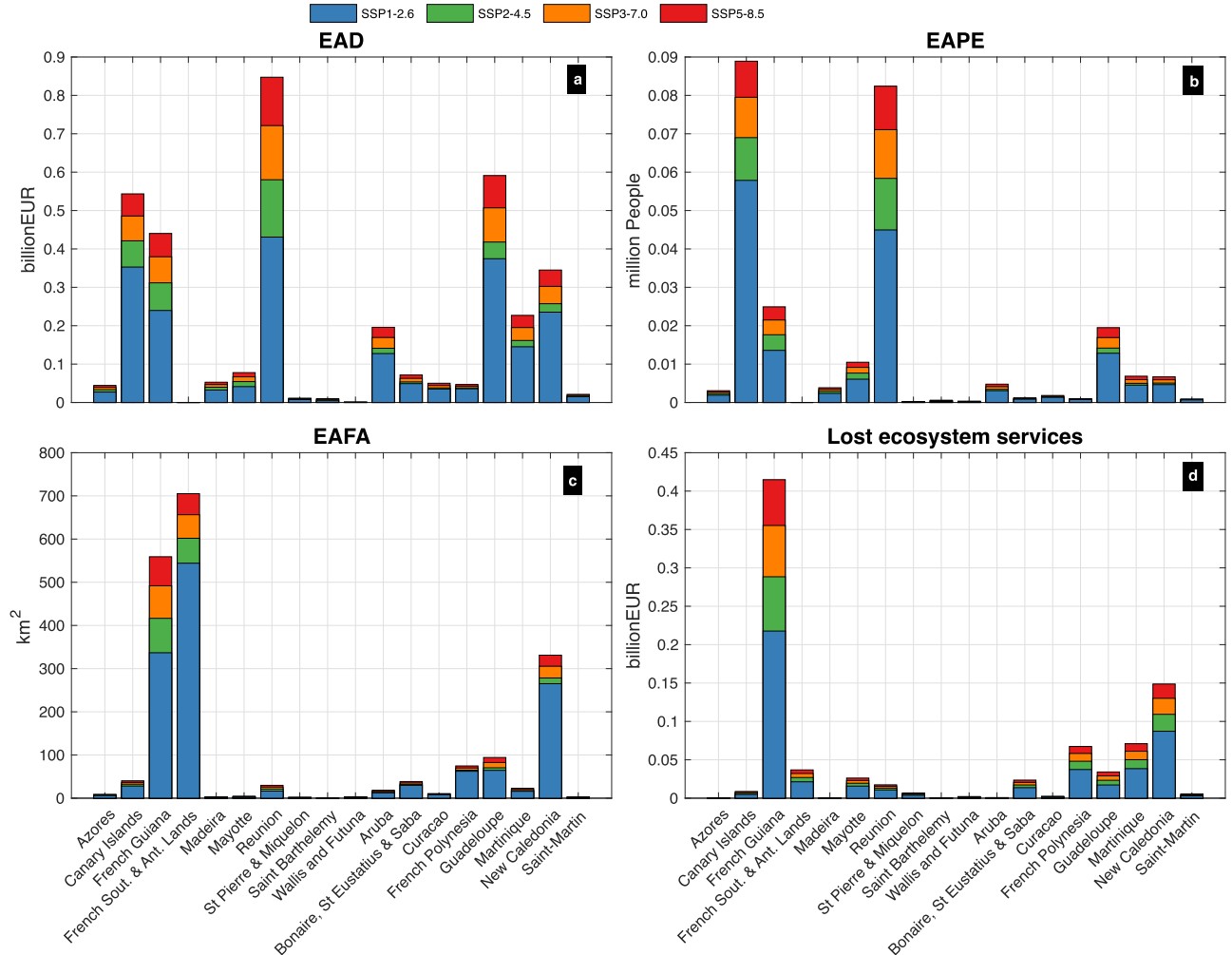

**Fig. 4 | Island-level expected annual values of coastal impacts along Europe's OCTs and ORs by the year 2100.** Values shown under all four scenarios ('low-emissions' (SSP1-2.6), 'moderate-emissions' (SSP2-4.5), 'high-emissions' (SSP3-7.0) and 'very-high-emissions' (SSP5-8.5)). EAD (**a**), expected annual number of people exposed (**b**), expected annual number of FA (**c**), and expected annual value of lost ecosystem services (**d**). The bars for each region are grouped into four stacks corresponding to median values for the present, 2050, 2100 and 2150, and the colours express the four scenarios.

are also the countries where the highest increase is projected, followed by Martinique (85.0–133.5), Aruba (71.0–109.7), and Curacao (49–70.5). Overall, the above rankings extend also to 2150.

## Anticipated loss of coastal ecosystem services

The current coastal ecosystem services in OCT/ORs are estimated at around 147.9 € billion annually, half of which comes from New Caledonia and 25% from French Guiana and French Polynesia (Fig. 3e). More than 90% of the ecosystem services are from tropical tree cover. Rising seas are projected to drive shoreline retreat, which will deplete a part of such ecosystem services. An estimated $198.35 \pm [20.24\text{–}435.04]$, $214.49 \pm [33.92\text{–}451.33]$, $222.41 \pm [37.21\text{–}461.86]$, and $245.24 \pm [47.57\text{–}492.98]$ € million of such ecosystem services are projected to be lost by the year 2050, under SSP1-2.6, SSP2-4.5, SSP3-7.0 and SSP5-8.5, respectively (Fig. 2d). The same estimates climb to $474.80 \pm [38.77\text{–}1053.49]$, $615.64 \pm [164.19\text{–}1250.29]$, $748.72 \pm [268.68\text{–}1434.20]$, and $867.60 \pm [366.63\text{–}1626.72]$ € million, respectively, by the end of the century. The above estimates imply emission mitigation benefits of at least 38.18% for the entire most likely range of the projections. By the year 2150, eco-system services of $736.41 \pm [20.58\text{–}1702.10]$, $1020.10 \pm [243.47\text{–}2131.25]$, $1314.44 \pm [445.63\text{–}2574.21]$, and $1499.61 \pm [571.23\text{–}2970.29]$ € million are projected to be lost, under SSP1-2.6, SSP2-4.5, SSP3-7.0 and SSP5-8.5, respectively (Fig. 2d).

By the years 2100 and 2150, the highest contributions come from the ORs (64.90%–67.58%, depending on the scenario), followed by the OCTs (32.42%–35.10%). By the end of the century, the corresponding median values are 308.15–578.47, and 166.65–289.13 € million, respectively (Fig. 3f), and by the year 2150, these values climb to 488.20–1013.47, and 248.21–486.14 € million, respectively. The country with the most losses in coastal ecosystem services by the end of the century is projected to be French Guiana (45.85%–47.82% of the total, range expresses SSP uncertainty), characterized by a long, low-lying coastline (Fig. 4d). It is followed by the New Caledonia (17.1%–18.4%), Martinique (8.1%–8.2%), French Polynesia (7.8%–7.9%), and French Southern and Antarctic Lands (4.2%–4.5%), respectively. The corresponding median lost ecosystem services values are 217.70–414.87 € million, 87.17–148.72 € million, 38.47–70.98 € million, 37.33–67.37 € million, and 21.35–36.83 € million, respectively (Fig. 3f). The corresponding 2150 median values are 349.02–732.13 € million, 128.16–247.75 € million, 60.13–123.29 € million, 57.34–115.71 € million, and 31.66–61.74 € million, respectively.

## Discussion

Our results show a strong increase in impacts from rising seas for Europe's OCTs and ORs. In the short term, the effect of climate mitigation is limited, and the impacts of high emissions in 2050 are

marginally worse compared to implementing stringent climate mitigation policies. The benefits of mitigation become more substantial over time. By 2100, our findings indicate an additional 139 km² of land lost to erosion, 102,774 more people exposed to floods and 1542 € billion of additional direct flood losses for the highest emissions scenario compared to the lowest emissions scenario. Such deviations are projected to keep increasing beyond the present century as sea levels will keep rising long after emissions have been reduced. Therefore, even under strong mitigation pathways such as SSP1-1.9, for which the maximum global average temperature is expected to occur before 2100[16], coastal impacts of SLR will continue to grow in time (at least until 2150 according to our analysis). For the lowest emission scenario (SSP1-1.9), and assuming present socioeconomic exposure in our analysis, impacts by 2150 will be 40% larger compared to impacts in 2100.

Impacts of rising sea levels will vary strongly among OCTs and ORs. Even for the lowest emissions scenario and as early as 2050, several OCTs and ORs are projected to experience expected annual direct economic impacts that exceed 1% of their GDP (e.g. French Guiana, Saint Pierre and Miquelon, Aruba, Guadeloupe, Martinique and New Caledonia). As time proceeds and under higher emissions scenarios, direct economic impacts could rapidly grow to several percentage points (more than 10% in the extreme case of French Guiana). These values reflect longer-term average impacts, while actual extreme flood events and their impacts can constitute substantial shocks for any economy, without even considering indirect impacts, like business interruption and other spill-over effects[16].

OCTs and ORs have several common characteristics with Small Island Developing States (SIDS), with some areas included in both groups; e.g. Aruba, Bonaire, Sint Eustatius and Saba, Guadeloupe, Martinique, New Caledonia and Saint-Martin. Our findings show that while rising sea levels will pose serious challenges to the resilience of OCTs and ORs societies and economy, especially after the second half of the century, the situation will not be as critical as for some SIDS. Without adaptation measures, low-lying atolls like Kiribati, Tuvalu, Maldives and the Marshal Islands are expected to face annual economic shocks exceeding 10% of their GDP even by the 2050s[9]. These are far larger shocks than the ones projected for French Guiana and Guadeloupe, the most affected areas in the present study. This is justified by the fact that most OCTs and ORs are volcanic and therefore their steep terrains leave a lower percentage of area exposed to rising seas compared to many SIDS.

However, the above do not mean that the projected impacts are negligible. The most affected areas in OCTs and ORs are flatter, low-lying coastal parts characterised by high population density. This includes, for example, the cities of Las Palmas de Gran Canarias, Cayenne of French Guiana, Ponta Delgada in the Azores, Mamoudzou and Dzaoudzi of Mayotte, Saint-Denis and Saint-Pierre of Reunion, Mata-Utu of Wallis and Saint-Pierre, and the capital of Saint Pierre and Miquelon. In all these cities, the average elevation doesn't exceed 10 m, and even if large parts will not be directly affected by floods, the disruption of everyday life will be substantial due to the inundation of critical assets like ports, roads and energy infrastructure, among others. The resilience of the latter is a critical issue since such assets tend to be in flat, low-lying areas, given the absence of higher altitude plains, especially in volcanic islands. Several OCTs/ORs airports lie below 10 m of elevation and will experience increasing flood risk, even under moderate SLR scenarios, e.g. the Dzaoudzi Pamandzi International Airport—Mayotte, Flamingo International Airport—Bonaire, Faa'a International Airport—French Polynesia, Pointe-à-Pitre International Airport—Guadeloupe and Saint-Pierre Airport—Saint Pierre and Miquelon. Airports, together with ports, which are anyway near MSL, are the lifelines of islandic communities, either for economic activity by ensuring the access of tourists to the island, or the other way

around, by providing the inhabitants' access to critical supplies and health services, among others.

All the above indicate that most of the affected areas cannot be abandoned and will need to be protected, also because relocation, despite being a valid adaptation option[17], often tends to be the least popular among residents[18,19]. However, recognising this challenge is only the first step. Building resilience in the EU's ORs and OCTs requires a deep understanding of their specific vulnerabilities and a collaborative approach to adaptation planning. A wide range of adaptation strategies can be considered, from traditional coastal defences and early warning systems to nature-based solutions (NBS), which offer flexible, sustainable alternatives. For example, beach nourishment and land reclamation can be applied in many coastal contexts, aligning with principles from the EU Floods Directive, such as the establishment of setback zones[20]. Built marine infrastructure, such are artificial reefs, are also very relevant in supporting ecosystems while protecting the coast, given the fact that many of the study areas are exposed to energetic wave conditions[21]. More ecosystem-focused approaches, like coral reef rehabilitation[22], may be suitable for certain ORs and OCTs, although others, such as mangrove restoration, may be less applicable due to the rarity of these ecosystems in some regions.

In any case, given the projected levels of sea-level rise, impermeable water barriers will likely be essential to protect key settlements and critical infrastructure from flooding. These "hard" protection measures can be made more sustainable through hybrid solutions that incorporate NBS principles. Due to the remote nature of many ORs and OCTs, adaptation efforts often face practical and logistical challenges. Financial capacity also varies across territories. However, as integral parts of the EU, most ORs have access to European funding instruments, including cohesion and infrastructure funds. In addition, some territories may be eligible for international support mechanisms, including Loss and Damage finance.

## Methods
### Coastal flood risk modelling framework
We assess impacts from SLR and episodic flooding along coastlines during the 21st century, considering both permanent inundation from SLR and tides and episodic flooding from coastal extremes. The analysis is based on the modular framework LISCOAST (Large-scale Integrated Sea-level and Coastal Assessment Tool). It combines state-of-the-art large-scale modelling tools and datasets to quantify hazard, exposure and vulnerability in coastal areas and compute consequent risks[4]. We consider the five principal Shared Socio-economic Pathway scenarios that span a range from ambitious mitigation to no emission policies: low-emissions (SSP1-2.6), reaching net-zero emissions after 2050 and achieving the Paris Agreement goal of holding the increase in global temperature to below 2 °C compared to pre-industrial levels; 'moderate emissions' (SSP2-4.5), implying stable emissions until mid-century, when they start to be reduced without reaching net-zero; high emissions (SSP3-7.0), with emissions rising constantly to almost double from current levels by the end of the century; and a high fossil-fuel development world throughout the 21st century (SSP5-8.5)[23]. For each of these scenarios, we generate probabilistic projections of mean and ESLs that give rise to permanent inundation or episodic flooding and combine them with exposure and vulnerability to quantify economic losses. In addition, we produce projections of coastal erosion and assess how the latter will affect future ecosystem services.

More details on the different steps of the analysis are provided below and in the flow diagram (Fig. 5).

### Present-day ESLs
Coastal areas are exposed to rising MSL and episodic high sea levels under extreme atmospheric conditions. ESLs are driven by the combined effect of MSL, tides and water level fluctuations due to waves and storm surges. We derive the contribution of each of these drivers

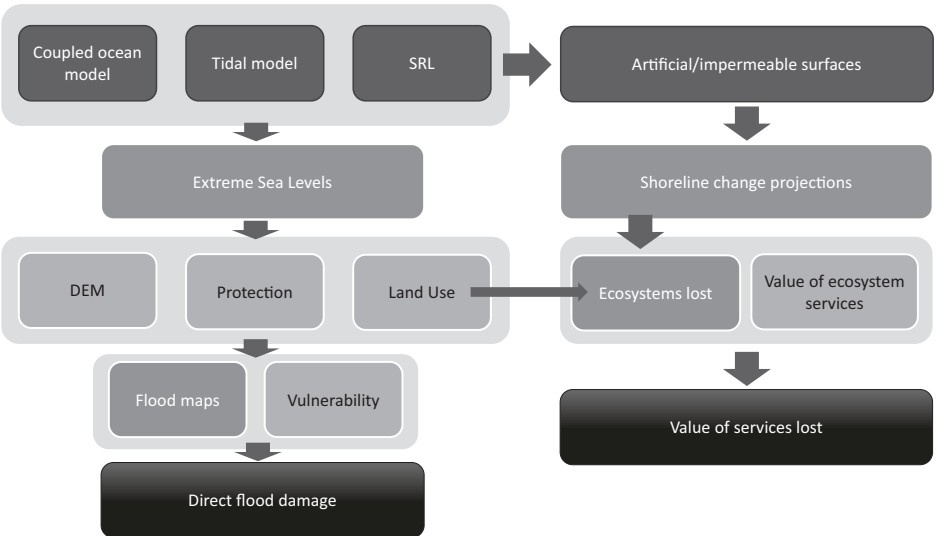

**Fig. 5 | Overview of the applied methodology.** Flow diagram of the input data and models to obtain intermediate data (e.g. ESLs, Shoreline Change Projections, Ecosystems Lost and Flood Maps) and the final outputs (e.g. estimates of direct flood losses and value of lost ecosystem services).

with state-of-the-art modelling tools and datasets and combine them to obtain ESLs every 1 km along the coastline.

For the baseline period, spanning from 1980 until 2020, we run a reanalysis of waves and storm surges based on a two-way coupled ocean model using an unstructured grid with a resolution ranging from ~50 km offshore to ~2 km nearshore. The coupled model system includes the Semi-implicit Cross-scale Hydroscience Integrated System Model (SCHISM)[24], configured in its two-dimensional barotropic mode and the 3rd-generation spectral wave model (WWM-V)[25]. The model accounts for the combined effects of wind, atmospheric pressure gradients, and tides. Bathymetric data are available from the European Marine Observation and Data Network (EMODnet) in angular coordinates at a resolution of 1/8 arc-minute (0.0021° of latitude and longitude; http://www.emodnet.eu/bathymetry) and are interpolated onto the computational grid.

We apply the coupled model to produce a reanalysis of waves and storm surges, forced by sea level pressure and wind speed data from ERA5[26]. The reanalysis is carried out without tidal forcing in order to make sure that our hindcast isolates the weather-driven component of ESLs from the tidal variations. Further details about the model setup and the validation can be found in Mentaschi et al.[27]. Since it is known that non-linear interactions between tides, waves and storm surges can be important in some areas, we apply a correction for the above effects following an approach similar to Arns[28]. To that end, we run a shorter 10-year reanalysis including tidal forces, and from the overlapping time series, we use copulas to produce a correction function for non-linear tidal effects on the water level anomaly and the significant wave height.

In order to improve the accuracy of the reanalysis data, we implement some additional steps as detailed below. Using satellite altimetry data, we apply a Quantile Mapping Bias Correction on both the water level anomaly and the significant wave height. This is done after compiling all coincident model and satellite values along 10 × 10 cells. To further improve the cyclone-related storm surge estimates, which have not been sufficiently resolved by our reanalysis, we did additional simulations of tropical cyclone-driven sea level anomalies using the Delft3D-FM model[29] forced by the IBTrACS best-track archive[30]. The reanalysis values are corrected by considering the tropical cyclone runs values when they are higher than those of our ERA5 runs. More information about the approach and data can be found in Vousdoukas et al.[9,31].

Spectral wave parameters provide one characteristic estimate for wave height, direction and period from the whole spectrum and therefore lack the detail needed to describe wave processes along complex shorelines. To overcome this shortcoming, we use the spectral peaks from the WWM-V model output and propagate each peak along a global transect dataset with 1 km alongshore resolution. The transect dataset includes information on the shoreline position, orientation, submerged and subaerial slope, among others. Details of the data and methods used to generate transects are provided in Athanasiou et al.[32]. We benefit from the complete spectral information from the wave model to propagate each peak at each time stamp along its corresponding transect using Snell's law[33]. We then estimate the wave breaking height by combining the peak wave parameters with the submerged profile slope. Subsequently, we obtain the wave run-up height $R_2$ based on the Stockdon empirical formula[34], after combining the breaking wave height and period with the subaerial beach profile slope. The above steps result in wave runup height estimates for each spectral peak, and we consider the highest value as the characteristic for the specific time stamp. We then combine the wave runup with the storm surge to obtain the meteorological tide and apply non-stationary extreme value analysis[35] to the time series to obtain estimates for different return periods.

Baseline ESLs are produced by combining the final meteorological tide time series with tidal elevations obtained from the FES2014 model[36]. Following the approach of Vousdoukas et al.[31], the high tide water level is considered, taking into account the range due to the spring-neap tide cycle.

### SLR projections
Relative SLR projections are obtained from the latest IPCC AR6 assessment[37–39] and incorporate the effects of the various components of future SLR, as simulated by climate models from the Coupled Model Intercomparison Project phase 6 (CMIP6), including steric SLR, dynamic sea-level change, contributions from glaciers and ice-caps, land-water storage and Glacial Isostatic Adjustment, among others.

### Projections of ESLs up to 2100
All ESL components (RSLR, tide, surges and $R_2$) are expressed as probability density functions (PDFs) that account for the different sources of uncertainty, and they are combined through Monte Carlo simulations in order to generate estimates of ESLs for the SSP

scenarios in each coastal segment (1 km alongshore resolution). Non-stationary extreme value analysis[35] is then applied to obtain for a range of return periods (i.e. 1, 2, 5, 10, 20, 50, 100, 200, 500, 1000, and 5000 years) PDFs of the corresponding return values of ESL throughout this century.

## Coastal flooding

We use hydraulic 2-D simulations along the entire coastline to estimate inundation extent and depth, following the approach presented by Vousdoukas et al.[40], running the Lisflood-ACC (LFP) model[41] at 30 m spatial resolution, using the estimated ESLs as forcing and considering hydraulic roughness derived from land-use maps[42]. Up to high-tide water levels (i.e. combination of MSL and high-tide), we apply the bathtub approach, and land below this seawater level and the corresponding assets are considered permanently inundated due to the effects of SLR. For episodic flooding, Liscflood-ACC is applied for each coastal segment with the model domain extending up to 200 km landwards in order to ensure the inclusion of all potentially hydrologically connected areas that may lie inland and away from the coast. The flood simulations are based on the recently published GLO-30 DEM[43], which is based on Synthetic Aperture Radar measurements and known to reduce the vertical bias of SRTM-based products. In addition, we apply post-processing using global LIDAR observations to further remove vertical bias, correcting for buildings and vegetation.

## Exposure and vulnerability

The resulting flood inundation maps are combined with exposure and vulnerability information to estimate population exposure and direct flood damages. For today's population exposure, we overlay the present inundation maps with the WorldPop 2020 population dataset (www.worldpop.org), which is an open and high-resolution geospatial dataset of population and demographic dynamics, with a focus on low and middle-income countries. The vulnerability to flooding is expressed through depth-damage functions (DDFs)[44], which define the relation between direct damage and flood inundation depth for different land use classes. Asset values are further scaled according to the GDP per capita available at 5 arc-minute resolution[45] in order to account for differences in the spatial distribution of wealth within countries. Baseline global land cover is available from the European Space Agency[46] at 10 m resolution. Given that more than 95% of the damages relate to built-up areas, the land use is corrected to take into account 30 m resolution gridded information on Global Human Built-up And Settlement Extent[47] (reference year 2010).

## Table 2 | Ecosystem service values (in 2022 euros at EU27 price level per hectare per year) according to ESA WorldCover class

| ESA WorldCover class | ESVD land cover type | Value/year/ha |
|---|---|---|
| Tree cover | Tropical forests | 82,638 |
| Tree cover | Temperate forests | 3736 |
| Shrubland | Woodland and shrubland | 534 |
| Grassland | Grassland | 1108 |
| Cropland | Cultivated areas | 5570 |
| Built-up | - | 0 |
| Bare/sparse vegetation | Inland un- or sparsely vegetated | 234 |
| Snow and ice | Inland un- or sparsely vegetated | 234 |
| Permanent water bodies | Rivers and lakes | 75,202 |
| Herbaceous wetland | Inland wetlands | 33,761 |
| Mangroves | Mangroves | 54,167 |
| Moss and lichen | Tundra | 568 |

## Risk assessment

For each coastal segment, the area flooded, number of people affected and direct flood losses are calculated at ~100 m resolution by combining flood inundation estimates with population and land use maps and the vulnerability functions. For areas that are inundated on a regular basis (which could happen in the future with SLR), defined as lying below the present high tide water level, assets are considered fully damaged, and the maximum loss according to the DDFs is applied. For areas inundated only during extreme events, the damage is estimated by applying the DDFs combined with the simulated inundation depth and land use information.

MSLs and ESLs, and the corresponding flood depths, are available as PDFs for different return periods (RP = 1, 2, 5, 10, 20, 50, 100, 200, 500, 1000, and 5000 years). Consequently, for each coastal segment, we obtain probabilistic estimates of flooded area (FA), population exposed (PE) and impact (D) from 1981 up to 2100. Integrating FA, PE and D over the return periods allows obtaining the EAFA, EAPE and direct economic Damage (EAD). We present and discuss our results about risk at global, regional, as well as country levels, and we focus on the median, 5th and 95th percentiles (very likely range).

## Land cover and ecosystem valuation

We use land cover data from the WorldCover 2020 dataset from the European Space Agency[46]. The dataset is an algorithmic classification of Sentinel-2 and Sentinel-1 images, identifying 10 land cover classes at 10 m resolution with a 74.4% overall accuracy[48]. We calculate the value of each land cover class using mean ecosystem value per year per hectare from the Ecosystem Services Valuation Database (ESVD), as presented by de Groot et al.[49]. The original values are in 2020 US dollars, which was first updated to 2022 dollars using the GDP deflator for the United States, and then converted to Euro at average price level of the European Union (27 countries) using the purchasing power parity conversion rates for 2022 (Eurostat, https://ec.europa.eu/eurostat). ESVD uses a different land cover classification, therefore, it is assigned to the ESA dataset as shown in Table 2. ESVD differentiates between tropical and temperate forests, which is not explicitly shown in WorldCover 2020, hence we assume that all forests are tropical except those located in regions with a temperate climate, i.e. the Azores, Madeira, the Canary Islands, Saint Pierre and Miquelon, and Kerguelen Islands (which are the only part of the French Southern and Antarctic Lands analysed here). The total value of ecosystem services is the value of all inland land cover types located within a 10 km radius from the coastline.

## Erosion extent

Projections of shoreline change are produced according to the methodology of Vousdoukas et al.[2], using the latest IPCC SLR projections, as in the flood risk analysis. The methodology combines estimates of ambient change and SLR retreat in a probabilistic framework, with the former assessed based on extrapolation of historical behaviour detected from satellites and the latter by using a modified version of the Bruun Rule. Detailed description of the approach and data used can be found in Vousdoukas et al.[2]. Erosion magnitude at different locations is connected with the base coastline derived from the ESA World Cover. The coastline is smoothed and split into detailed segments with an average length of 30 m. Each segment of the coastline is assigned a sector of the inland land cover based on nearest-neighbour analysis. However, as unlimited backshore space for shoreline retreat is often not a realistic assumption due to topography or human coastline protection activities, we constrain the erosion extent by introducing two types of barriers into the analysis in a similar manner to Paprotny et al.[50]. Firstly, artificial barriers representing various anthropogenic structures are extracted from the GHS built-up surface (R2023) dataset for reference year 2018 at 10-m resolution[51], which is part of the Global Human Settlement Layer database. Any occurrence

of built-up surfaces in a grid cell is considered a barrier to erosion. Second, topographic barriers representing high ground that is unlikely to be easily eroded are derived from DeltaDTM[43]. Each coastal sector corresponding to a 30-m (on average) section of the coastline is intersected with the barriers, erasing areas that are vulnerable to erosion. Parts of the sector that are no longer connected directly to the coastline were removed. The resulting layer of areas where erosion is possible is used to clip the buffer around each coastline segment generated according to the erosion extent per scenario in a given location. Finally, the constrained erosion layer is intersected with the land cover dataset, and the value of ecosystem services is calculated according to the eroded area per land cover class.

## Limitations

Coastal change patterns can modulate flood events, since the bathymetry affects nearshore waves and storm surges and thus ESLs[52]. As a result, detailed information on nearshore bathymetry and modelling of the latter's interactions with waves and currents is essential for reliable estimations of ESLs and thus flooding. As a result, the ideal framework should combine dynamic assessments of the coastal morphological change and flood risk, however, such an effort would be on the edge, if not beyond the current state of the art. In addition, the scale of the present study and the available datasets do not permit such detailed modelling. The above implies that one of the inevitable limitations of this study is that we do not directly resolve the interactions between the two hazards.

Coastal erosion is projected to be mainly driven by SLR, in contrast to floods, for which meteorological and astronomical tides are the main drivers. More than 95% of the coastal flood losses come from artificial areas, which also act as the landward limit of coastal erosion. This means that in case of shoreline retreat in front of urbanised areas, the double counting of the economic damages from floods and lost ecosystem services is very small, since the second number is much higher than the first (retreat takes place only in non-artificial areas where flood losses are very small). On the other hand, in natural coastlines, shoreline retreat will continue unimpeded, resulting in even higher losses from ecosystem services. But also, in this case, coastal flood losses are less significant due to the lack of built areas. One unresolved issue remains the fact that in the case of shoreline accretion in front of artificial coastlines, the additional land will provide natural protection, which will reduce the amount of FAs backshore. But such cases of shoreline accretion in front of urbanisation are rarely natural but are a result of human intervention, which implies also shoreward expansion of the built area. Given all the above, and considering the scope of the work, we consider that the nonlinear interactions are insignificant compared to the other sources of uncertainty.

Our analysis is affected by several unavoidable sources of uncertainty, such as the vertical accuracy of the digital elevation model, the ocean model errors, the expert-elicited vulnerability, and the protection standards, among others, all known and extensively discussed in the literature[53–55]. Such previous studies have shown that the highest uncertainty lies in the translation of floods and erosion to economic impacts[54,56] and in predicting human behaviour which affects future greenhouse gas emissions, socioeconomic growth and the ambient change of the coasts through coastal management. In our framework, we try to minimise to the highest possible extent such sources of uncertainty and quantify the residual uncertainty, expressing our output as percentiles and discussing the very likely range.

While our analysis accounts for uncertainty in the physical hazard chain, it is also important to consider sources of uncertainty in the economic baseline used for comparison, namely GDP. Regional differences in financial capacity, institutional readiness, and long-term growth trajectories can introduce substantial variability in GDP projections, complicating cross-regional comparisons of damages relative to economic output. In addition, macroeconomic impacts of climate change are highly sensitive to assumptions about whether climate shocks affect GDP levels or long-term growth rates[57]. Although our results focus on relative damages as a share of GDP, readers should be aware that these estimates remain sensitive to broader economic uncertainties not fully captured in the physical impact modelling. Also, our damage estimates assume a no-adaptation scenario and thus they represent a stylised, increasingly implausible reference case intended to isolate potential physical impacts. They are meaningful as they can allow estimating the cost of inaction, but should be interpreted as a potential upper bound of damages[58]. In the absence of more detailed information, we assume that the value of the ecosystem does not vary spatially. This limitation may be inevitable, but it is one of the many sources of epistemic uncertainty related to quantifying the economic benefits from ecosystems[59–62]. More spatially explicit valuation methods, such as benefit transfer functions that adjust for local context[62], could improve accuracy in future work. Additionally, the marginal value of ecosystem services may increase non-linearly with both rising scarcity and growing wealth[59,63]. This suggests that the long-term value of threatened coastal ecosystems may be underestimated in static valuation approaches. Another limitation of our approach is that, given the scarcity of such information, we are not considering any uncertainty in the ecosystem service value estimates. Acknowledging all the above, we emphasise that our results are intended to signal the potential magnitude of ecosystem service loss under continued coastal change.

## Ethics and inclusion statement

The authors declare no ethics issues such as research on race, sex, ethnicity, clinical trials or humans or animals. The list of authors is broad, while local and regional studies have been considered.

## Data availability

The models and datasets presented are part of the integrated risk assessment tool LISCoAsT (large-scale integrated sea-level and coastal assessment tool) developed by the Joint Research Centre of the European Commission. All data used are open access, and links are provided, while all source data are provided in the Supplementary Dataset 1.

## Code availability

Most of the code that supported the findings of this study is already open access with references provided in the manuscript; specific tools which are not available in public repositories will be available on reasonable request from the corresponding authors.

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

## Acknowledgements

We thank the sea-level projection authors for developing and making the sea-level rise projections available, multiple funding agencies for supporting the development of the projections, and the NASA sea-level change team for developing and hosting the IPCC AR6 Sea-Level Projection Tool.

## Author contributions

M.I.V., L.M., D.P., and L.F.: conceptualisation. M.I.V., D.P., L.M., and I.M.: preliminary and exploratory analysis. M.I.V., D.P., and L.F.: methodology. All authors: validation. M.I.V. and D.P.: formal analysis. M.I.V. and D.P.: investigation. M.I.V., D.P., and L.F.: resources. M.I.V., L.M., D.P., and I.M.: data integration. M.I.V., D.P., I.M., and L.F.: writing—original draft. All authors: writing—review and editing. M.I.V.: visualisation. The information and views set out are those of the author(s) only and should not be considered as representative of the European Commission's official position.

## Competing interests

The authors declare no competing interests.
