## [Transparent Peer Review file · Nature Communications]

Coastal flood impacts and lost ecosystem services along Europe's Outermost Regions and Overseas Countries and Territories

Corresponding Author: Dr Michalis Vousdoukas

Version 0:

Reviewer comments:

Reviewer #1

(Remarks to the Author)

The manuscript addresses coastal flood impacts and the loss of ecosystem services in Europe's Outermost Regions and Overseas Countries and Territories.

It investigates areas that are poorly studied in terms of flooding, submergence of flat regions, and coastal erosion along coastlines.

The manuscript is rich in useful information, presents a very detailed methodology, and can serve as a guideline for conducting similar analyses in comparable contexts.

For these reasons, I recommend its publication in Nature Communications after minor revisions.

Minor comments:

Introduction: I suggest including a figure that shows the Outermost Regions and Overseas Countries and Territories analyzed in the article.

Detailed methods: A flow diagram illustrating the applied methodology could help readers reproduce the analysis.

Figure 1: Please review the caption for Figure 1.

Discussion: Could the results be grouped by location? This grouping might help identify geographical areas most affected by coastal risks or verify if there are regions with common characteristics. In this regard, I encourage a more detailed discussion on this aspect.

Reviewer #2

(Remarks to the Author)

Key Results

The manuscript provides a detailed, quantitative analysis of the socio-economic and ecological impacts of sea-level rise and coastal flooding on Europe's Outermost Regions (ORs) and Overseas Countries and Territories (OCTs). Using a state-of-the-art data-modeling framework, it highlights the significant vulnerabilities of these regions, projecting future damages under multiple emissions scenarios and underscoring the critical importance of mitigation and adaptation strategies.

Major Comments

1. Significance and Context

Strengths:

The manuscript addresses an underrepresented research gap, offering a novel focus on ORs and OCTs. It employs cutting-edge methodologies to quantify impacts in these regions, contributing valuable insights into their vulnerabilities.

Suggestions:

a) Expand the discussion to connect the findings with other vulnerable regions, such as Small Island Developing States (SIDS). For example, compare projected GDP losses, population displacement, or ecosystem service impacts between

ORs/OCTs and SIDS.

b) Cite recent studies that emphasize island-specific climate change vulnerabilities to situate the work within a broader research context.

2. Methodological Soundness

Strengths:

The manuscript utilizes robust models (e.g., LISCOAST) and credible data sources (e.g., IPCC AR6, ESA WorldCover). Limitations, such as decoupling of erosion and flooding processes, are acknowledged.

Suggestions:

a) Explicitly discuss the potential biases introduced by treating erosion and flooding separately. For example, would integrating these processes amplify or mitigate the projected impacts? A brief qualitative analysis in the discussion would strengthen the methodology's transparency.

b) Consider providing a supplementary appendix with simplified flowcharts or diagrams of the methodological framework to help readers navigate complex processes like hydrodynamic modeling and vulnerability assessments.

3. Policy Relevance and Adaptation Strategies

Strengths:

The manuscript mentions nature-based solutions (NBS) as a critical tool for adaptation.

Suggestions:

a) Include concrete examples of NBS tailored to the studied regions. For example, mangrove restoration in tropical areas, coral reef rehabilitation, or managed retreat strategies for low-lying atolls.

b) Discuss potential barriers to implementing these solutions in ORs and OCTs (e.g., financial, governance, or technical challenges) and suggest strategies to overcome them.

c) Relate adaptation strategies to policy frameworks in the EU and other governing bodies of OCTs/ORs to make the findings actionable for stakeholders.

4. Data Transparency and Reproducibility

Strengths: The manuscript uses extensive datasets and robust modeling frameworks.

Suggestions:

a) Clearly state where datasets and model codes are available for public access. If sharing full data is infeasible, consider releasing derived datasets or visualizations of model outputs to enhance transparency.

b) Suggest repositories such as Zenodo or Figshare for hosting key datasets and codes, aligning with open science practices.

5. Data Presentation and Figures

Strengths: Figures are informative and well-designed for specialists.

Suggestions:

a) Simplify complex figures like Figures 1 and 2 by focusing on representative scenarios (e.g., SSP1-2.6 and SSP5-8.5) or regions. Figure 1 and 2 are very hard to digest at the moment.

b) Provide more intuitive legends and labels. For example, use consistent color coding across scenarios and add arrows or annotations to highlight key trends.

c) Add a supplementary table summarizing key numerical findings (e.g., GDP losses, population exposure, ecosystem service values) for each region. This will make the data more accessible for policymakers and non-specialist audiences.

Minor Comments

1. Abstract: Include a specific statement on the mitigation benefits, such as: "Under SSP1-2.6, economic damages could be reduced by X%, highlighting the importance of stringent emissions reductions."

2. Introduction: Add a brief paragraph contrasting ORs and OCTs with continental regions in terms of vulnerabilities to sea-level rise and flooding.

3. Language and Jargon: While the manuscript is generally well-written, some sections (e.g., "Detailed Methods") are dense with technical jargon. Simplify descriptions where possible, such as rephrasing "probabilistic extrapolation of historical

behavior detected from satellites” to “using historical satellite data to estimate future shoreline changes.” Give the manuscript a little rest and then with fresh eyes, try to simplify the use of language as much as possible.

4. References: The reference list is comprehensive, but it could include more studies on successful adaptation strategies in island contexts.

5. Adaptation Strategies: Add a dedicated subsection or paragraph on adaptation strategies with specific examples and their feasibility for ORs and OCTs.

6. Uncertainty and Scope: Expand the discussion on uncertainty. For example, discuss how socio-economic variability (e.g., migration patterns, GDP growth) might influence the projections. You might want to address limitations of the decoupled flooding and erosion analysis by outlining potential pathways for integrating these processes in future studies.

7. Accessibility: You might wish to include a "Key Findings" section summarizing the main results in plain language for policymakers and stakeholders where you best see fit.

8. Figures and Tables: Redesign Figure 2 to highlight mitigation benefits across scenarios more clearly. For example, use bar charts to show absolute differences in projected damages under SSP1-2.6 versus SSP5-8.5.

Overall Recommendation

The manuscript is a significant contribution to climate change research and provides essential insights into the vulnerabilities of ORs and OCTs. While the study is methodologically sound and highly relevant, revisions are needed to enhance clarity, accessibility, and policy relevance. These improvements will ensure the manuscript meets the high standards of Nature Communications and achieves maximum impact in the scientific and policy communities.

Version 1:

Reviewer comments:

Reviewer #3

(Remarks to the Author)

Key Results

This study investigates coastal flooding impacts and the subsequent change in expected annual damages and ecosystem services in Europe's Outermost Regions and Overseas Countries and Territories. It investigates areas that are poorly studied but are disproportionately impacted by changing climatic conditions. The paper is dense, providing a very high level of detail. This makes it harder to read but allows the reader to extract more information about changes across very large and spatially diverse areas. Much of the methodology is detailed and the study does an excellent job highlighting the significant vulnerabilities in these regions. It also clearly communicates the importance of adaptation in the areas.

Major Comments

Previous Efforts to address previous reviewer comments: I think the efforts to address the first round of comments were mostly sufficient. I added additional comments related to estimation and interpretation of economic results.

Significance and Context

This study represents an important topic in highly vulnerable regions (OOs & OCTs). Many of the methodologies are cutting edge. The economic analyses seems to lag the other analysis, especially as it relates to ecosystem service valuation and the comparison of expected damages to GDP over time.

Uncertainty in Economic Results

The manuscript seems insufficient in accounting for uncertainty associated with the economic results. My biggest issue is that this study spends a significant amount of effort discussing the economic implications of change within and outside of markets, but it does not adequately communicate how uncertainty manifests itself in the estimates. This could be more explicit in the paper. The literature related to economic uncertainty seems insufficient and largely ignores uncertainty in the non-market context (i.e. ecosystem services). In addition, damages are reported relative to GDP using a baseline that appears to omit location-specific adaptation and to treat GDP similarly across locations applying the different risk profiles but without accounting for heterogeneous economic growth or adaptive capacity.

Assessment of Uncertainty in Changes in Damage Relative to GDP: The analysis is excellent in accounting for potential uncertainty in the physical hazard chain related to exposure, vulnerability, and damages. The literature cited also shows the importance of adaptation and the study has extensive discussion of adaptation following the results. It seems that the manuscript can expand on the description of the uncertainty in the economic results and what drives that uncertainty. For example, Bachner et al (2022) appears to be relevant to this application. Bachner et al. (2022) seem to describe a no adaptation scenario as an implausible reference point when forecasting damage and GDP changes from sea level rise into the future. It is important to then consider how a no-adaptation dollar estimate can be interpreted since these areas will be forced to adapt out of necessity. This likely has implications for how a reader should interpret the economic results, with damage estimates representing potential upper bounds if realistic adaptation is not accounted for. Further, economic differences across locations are likely to lead to different rates of GDP growth, which could be significant over a long time horizon. That makes comparisons between GDP and expected damages less certain between locations over the long time horizon. Newell et al (2021) also provides relevant background information for explaining potential model uncertainties.

Newell investigates many plausible models linking temperature and GDP. While temperature impacts are different than damages from sea level rise and erosion, that work discusses both level effects and growth effects on GDP. Newell results show estimates of GDP changes to be uncertain because growth responses are sensitive to modeling assumptions. Since this paper compares damage estimates to GDP, this seems like a relevant topic, even if it is only referred to in the detailed methodology as a limitation and area of future research. The manuscript already mentions differences in financial capacity across locations in the discussion, so this seems like a natural extension that acknowledges the complexity of economic outcomes when faced with these risks.

Assessment of Ecosystem Services

In the paper, the manuscript first provides an estimate for, and then highlights changes in, aggregate ecosystem service value from changing coastal conditions. Costanza et al. (2014) argue that for raising awareness, Total Aggregate Values applied to regional or global scales are appropriate and low precision is acceptable. I find this a reasonable argument, but the paper should be explicit in communicating this so not to mislead readers or overstate its policy applications beyond highlighting value at risk. I do have some concerns that the manuscript uses global mean values for ecosystem services rather than following some type of transfer values that are adjusted--possibly even values using a more spatially explicit transfer function that accounts for differences in sites and populations. Johnston et al. (2021) provide details into best practices for benefit transfer. The manuscript makes no note of the assumptions necessary for the transfer of mean global values in this context or the implications of using these types of values. The manuscript really glosses over the limitations of those estimates.

Additionally, the manuscript does not provide estimates of uncertainty global mean values for ecosystem services. It could still rely on mean values in analysis while also communicating variability using standard errors or confidence intervals. Fenichel and Abbot (2014) argue that as ecosystem services become scarcer over time, the marginal value of those services can increase non-linearly. Drupp et al. (2024) extend that argument. This highlights the importance of feedbacks between ecosystems and economic behavior within and outside markets. I think it is important for readers to realize that the marginal value of ecosystem services rises with both increased scarcity and increased wealth. You could add these types of references to the detailed methodology to communicate uncertainty over long time periods.

Policy Relevance

This is a very important topic for these impacted regions. There will need to be significant efforts for both protection and adaptation in the study areas. While I have concerns over how policy makers should interpret the economic estimates, they provide an excellent opportunity to raise awareness. The manuscript can both clearly state the uncertainties and the extensive assets at risk.

Minor Comments:

- On line 81-82, you state that these “flooded areas do not always translate into population exposure and economic impacts.” I am assuming that you are referring to market based economic impacts. This is slightly confusing since you also include ecosystem service valuations in the paper and ecosystem service values can manifest on local, regional, or global levels. This is very minor, but slight rewording may be helpful.
- Spelling: change asses to assess on line 69

Overall assessment

Overall, I found this to be a very strong paper, but it is inadequate in communicating the limitations of the economic results and in providing guidance for how a reader should interpret the economic results. I recommend the authors clarify these areas both in the main portion of the manuscript and in detailed methods. With those changes, this manuscript represents a strong contribution to the climate change literature.

My Expertise

I am an environmental economist.

Citations

- Bachner, G., Lincke, D., & Hinkel, J. (2022). The macroeconomic effects of adapting to high-end sea-level rise via protection and migration. *Nature Communications*, 13(1), 5705.
- Costanza, R., De Groot, R., Sutton, P., Van der Ploeg, S., Anderson, S. J., Kubiszewski, I., ... & Turner, R. K. (2014). Changes in the global value of ecosystem services. *Global environmental change*, 26, 152-158.
- Drupp, M. A., Hänsel, M. C., Fenichel, E. P., Freeman, M., Gollier, C., Groom, B., ... & Venmans, F. (2024). Accounting for the increasing benefits from scarce ecosystems. *Science*, 383(6687), 1062-1064.
- Fenichel, E. P., & Abbott, J. K. (2014). Natural capital: from metaphor to measurement. *Journal of the Association of Environmental and Resource Economists*, 1(1/2), 1-27.
- Johnston, R. J., Boyle, K. J., Loureiro, M. L., Navrud, S., & Rolfe, J. (2021). Guidance to enhance the validity and credibility of environmental benefit transfers. *Environmental and Resource Economics*, 79(3), 575-624.
- Newell, R. G., Prest, B. C., & Sexton, S. E. (2021). The GDP-temperature relationship: implications for climate change damages. *Journal of Environmental Economics and Management*, 108, 102445.

Version 2:

Reviewer comments:

Reviewer #3

(Remarks to the Author)

I have not further comments related to this manuscript. The authors addressed my comments by identifying those issues as limitations in their current analysis and communicate their primary economic results (relative damage as a share of GDP) as upper bounds. Similarly, the authors identify the limitations of ecosystem service valuation under these conditions. Given the primary focus of this paper by identifying the uncertainty in the physical hazard chain, I think it is reasonable to communicate other economic uncertainties as long as it is clear that people in these regions will adapt to changes and that ecosystem service values depend on local context and relative scarcity.

Overall, I am satisfied with the authors communicating the limitations of their economic analysis. I feel that this is an important topic in an understudied area.

Response to the Reviewers' comments

Authors: We would like to thank the reviewers and the editor for the positive comments. We have revised the manuscript according to the reviews and we discuss each point in the sections to follow. We would like to thank you again for the constructive effort and we are at your convenience for any additional changes needed.

Reviewer #1

The manuscript addresses coastal flood impacts and the loss of ecosystem services in Europe's Outermost Regions and Overseas Countries and Territories.

It investigates areas that are poorly studied in terms of flooding, submergence of flat regions, and coastal erosion along coastlines.

The manuscript is rich in useful information, presents a very detailed methodology, and can serve as a guideline for conducting similar analyses in comparable contexts.

For these reasons, I recommend its publication in Nature Communications after minor revisions.

Authors: We would like to thank the reviewer for the positive comments

Minor comments:

Introduction: I suggest including a figure that shows the Outermost Regions and Overseas Countries and Territories analyzed in the article.

Authors: We have added the map as suggested by the reviewer and it is now Figure 1.

Detailed methods: A flow diagram illustrating the applied methodology could help readers reproduce the analysis.

Authors: We have added such a flow diagram in Figure 5.

Figure 1: Please review the caption for Figure 1.

Authors: We have reviewed the caption and made several changes. We hope it now reads better.

Discussion: Could the results be grouped by location? This grouping might help identify geographical areas most affected by coastal risks or verify if there are regions with common characteristics. In this regard, I encourage a more detailed discussion on this aspect.

Authors: As shown in the map the reviewer suggested (Figure 1) the islands are quite scattered to be grouped geographically, while in addition they are characterized by distinct characteristics. This is a difference with SIDS which are more in number and form clear geographical groups. Still, we have expanded the discussion to several dimensions, as also suggested by both reviewers.

Reviewer #2 (Remarks to the Author):

Key Results

The manuscript provides a detailed, quantitative analysis of the socio-economic and ecological impacts of sea-level rise and coastal flooding on Europe's Outermost Regions (ORs) and Overseas Countries and Territories (OCTs). Using a state-of-the-art data-modeling framework, it highlights the significant vulnerabilities of these regions, projecting future damages under multiple emissions scenarios and underscoring the critical importance of mitigation and adaptation strategies.

Major Comments

1. Significance and Context

Strengths:

The manuscript addresses an underrepresented research gap, offering a novel focus on ORs and OCTs. It employs cutting-edge methodologies to quantify impacts in these regions, contributing valuable insights into their vulnerabilities.

Authors: We would like to thank the reviewer for the positive comments about our work. We have tried to improve our manuscript as much as possible following the suggestions.

Suggestions:

a) Expand the discussion to connect the findings with other vulnerable regions, such as Small Island Developing States (SIDS). For example, compare projected GDP losses, population displacement, or ecosystem service impacts between ORs/OCTs and SIDS.

Authors: We have expanded the discussion including comparisons with results for SIDS and thanks to the reviewer's comment we are now confident that the discussion is more lively and informative.

b) Cite recent studies that emphasize island-specific climate change vulnerabilities to situate the work within a broader research context.

Authors: Since we have expanded both the introduction and discussion, the manuscript cites several recent articles that touch SIDS, adaptation and other relevant issues.

2. Methodological Soundness

Strengths:

The manuscript utilizes robust models (e.g., LISCOAST) and credible data sources (e.g., IPCC AR6, ESA WorldCover). Limitations, such as decoupling of erosion and flooding processes, are acknowledged.

Suggestions:

a) Explicitly discuss the potential biases introduced by treating erosion and flooding separately. For example, would integrating these processes amplify or mitigate the projected impacts? A brief qualitative analysis in the discussion would strengthen the methodology's transparency.

Authors: We have a whole paragraph in the methods dedicated on that issue. There we discuss the necessity for that assumption and explain why we believe it doesn't affect the results to a significant level. We have expanded that section and we hope now it is clearer for the reader.

b) Consider providing a supplementary appendix with simplified flowcharts or diagrams of the methodological framework to help readers navigate complex processes like hydrodynamic modeling and vulnerability assessments.

Authors: We have added a flow diagram of the methodology as this was also a suggestion by Reviewer 1 (now figure 5).

3. Policy Relevance and Adaptation Strategies

Strengths:

The manuscript mentions nature-based solutions (NBS) as a critical tool for adaptation.

Suggestions:

a) Include concrete examples of NBS tailored to the studied regions. For example, mangrove restoration in tropical areas, coral reef rehabilitation, or managed retreat strategies for low-lying atolls.

b) Discuss potential barriers to implementing these solutions in ORs and OCTs (e.g., financial, governance, or technical challenges) and suggest strategies to overcome them.

c) Relate adaptation strategies to policy frameworks in the EU and other governing bodies of OCTs/ORs to make the findings actionable for stakeholders.

Authors: We agree with the reviewer that discussing further NBS strengthens the paper and we elaborate on this in the last paragraphs. We do so, at the same time avoiding proposing solutions and interfering with the political decision making process of adaptation, especially because some of the authors are part of the EC and have restrictions relative to that.

4. Data Transparency and Reproducibility

Strengths: The manuscript uses extensive datasets and robust modeling frameworks.

Suggestions:

a) Clearly state where datasets and model codes are available for public access. If sharing full data is infeasible, consider releasing derived datasets or visualizations of model outputs to enhance transparency.

b) Suggest repositories such as Zenodo or Figshare for hosting key datasets

and codes, aligning with open science practices.

Authors: Nature journals have a strict policy in open access data and open source tools which we fully comply with. We have received several forms to fill on these topics and the published manuscript will satisfy all data transparency and reproducibility criteria.

5. Data Presentation and Figures

Strengths: Figures are informative and well-designed for specialists.

Suggestions:

a) Simplify complex figures like Figures 1 and 2 by focusing on representative scenarios (e.g., SSP1-2.6 and SSP5-8.5) or regions. Figure 1 and 2 are very hard to digest at the moment.

Authors: We understand that especially figure 1 contains a lot of information, but looking at other published articles in NCOMMS it is not an extreme case. After all, most authors are obliged to put a lot of information in the limited number of figures allowed. Still, if the reviewer (or the editor) insists we are available to simplify figure 1 and move parts to the SI. From our side we believe that as the figure is cited several times in the manuscript, it would be more cumbersome for the reader to have to address to the SI, rather than having only one more complex figure in the main manuscript.

We don't think figure 2 is complicated and breaking it up in 4 figures would result in too many figures that would not be justifiable. In any case we have worked and improved the captions.

b) Provide more intuitive legends and labels. For example, use consistent color coding across scenarios and add arrows or annotations to highlight key trends.

Authors: The color coding of the scenarios is consistent among all figures. We provide eps versions of the figures so any discrepancies due to technical issues (like renderers) can be resolved by the journals editing team.

c) Add a supplementary table summarizing key numerical findings (e.g., GDP losses, population exposure, ecosystem service values) for each region. This

will make the data more accessible for policymakers and non-specialist audiences.

Authors: Tables of all variables are provided in the public dataset. We have added a simplified table in the manuscript according to the reviewer's comment (now Table 2).

Minor Comments

1. Abstract: Include a specific statement on the mitigation benefits, such as: "Under SSP1-2.6, economic damages could be reduced by X%, highlighting the importance of stringent emissions reductions."

Authors: Such statement is present in the abstract: 'Our study shows the increasing benefits in time of stringent climate mitigation, which could nearly halve these impacts in the long run'.

We feel that the way it is written is more appropriate to the style of the journal ('stringent climate mitigation'), than referring to actual SSP scenarios, but if the reviewer or the editor insists we can modify the sentence.

2. Introduction: Add a brief paragraph contrasting ORs and OCTs with continental regions in terms of vulnerabilities to sea-level rise and flooding.

Authors: We have expanded the first paragraph of the introduction according to the reviewer's comment. We now believe that such challenges are better presented not only in this, but also other parts of the introduction.

3. Language and Jargon: While the manuscript is generally well-written, some sections (e.g., "Detailed Methods") are dense with technical jargon. Simplify descriptions where possible, such as rephrasing "probabilistic extrapolation of historical behavior detected from satellites" to "using historical satellite data to estimate future shoreline changes." Give the manuscript a little rest and then with fresh eyes, try to simplify the use of language as much as possible.

Authors: We tried to do modifications to the proposed direction without having important technical details being lost. After all it is the Methods where experts may look for specific information, while we are confident that the main manuscript is written in a simpler, more pleasant style.

4. References: The reference list is comprehensive, but it could include more studies on successful adaptation strategies in island contexts.

Authors: We have added several references since we have expanded the introduction and discussion, according to the reviewers' comments.

5. Adaptation Strategies: Add a dedicated subsection or paragraph on adaptation strategies with specific examples and their feasibility for ORs and OCTs.

Authors: We have expanded the discussion to that direction, also according to the reviewer's comment 3c. At the same time we try to avoid to propose solutions and interfere with the political decision making process of adaptation, especially because some of the authors are part of the EC and have restrictions relative to that.

6. Uncertainty and Scope: Expand the discussion on uncertainty. For example, discuss how socio-economic variability (e.g., migration patterns, GDP growth) might influence the projections. You might want to address limitations of the decoupled flooding and erosion analysis by outlining potential pathways for integrating these processes in future studies.

Authors: We have added a section dedicated to uncertainty in the methods.

7. Accessibility: You might wish to include a "Key Findings" section summarizing the main results in plain language for policymakers and stakeholders where you best see fit.

Authors: We checked the journal's guidelines, and such a section is not recommended. We discern from the guidelines that the abstract should serve that purpose, but if the editor recommends adding a Key Findings section we will do it.

8. Figures and Tables: Redesign Figure 2 to highlight mitigation benefits across scenarios more clearly. For example, use bar charts to show absolute differences in projected damages under SSP1-2.6 versus SSP5-8.5.

Authors: We understand the nature of the suggestion, but we think that while changing the figure would make the 'benefits of mitigation'

message stronger, it would also come with disadvantages, e.g. not showing information of the temporal evolution of Impacts and differences among all scenarios, that some readers might be interested in. For that reason, we prefer to keep the figure as it is, since anyway such figures are ambiguous in IPCC reports and most readers are already familiar with them.

Overall Recommendation

The manuscript is a significant contribution to climate change research and provides essential insights into the vulnerabilities of ORs and OCTs. While the study is methodologically sound and highly relevant, revisions are needed to enhance clarity, accessibility, and policy relevance. These improvements will ensure the manuscript meets the high standards of Nature Communications and achieves maximum impact in the scientific and policy communities.

Authors: We would like to thank the reviewer for the constructive comments and we are also confident that the manuscript has improved substantially after implementing them.

REVIEWER COMMENTS

Reviewer #3 (Remarks to the Author):

Key Results

This study investigates coastal flooding impacts and the subsequent change in expected annual damages and ecosystem services in Europe's Outermost Regions and Overseas Countries and Territories. It investigates areas that are poorly studied but are disproportionately impacted by changing climatic conditions. The paper is dense, providing a very high level of detail. This makes it harder to read but allows the reader to extract more information about changes across very large and spatially diverse areas. Much of the methodology is detailed and the study does an excellent job highlighting the significant vulnerabilities in these regions. It also clearly communicates the importance of adaptation in the areas.

AUTHORS: We would like to thank the reviewer for his positive comments.

Major Comments

Previous Efforts to address previous reviewer comments: I think the efforts to address the first round of comments were mostly sufficient. I added additional comments related to estimation and interpretation of economic results.

Significance and Context

This study represents an important topic in highly vulnerable regions (OOs & OCTs). Many of the methodologies are cutting edge. The economic analyses seems to lag the other analysis, especially as it relates to ecosystem service valuation and the comparison of expected damages to GDP over time.

Uncertainty in Economic Results

The manuscript seems insufficient in accounting for uncertainty associated with the economic results. My biggest issue is that this study spends a significant amount of effort discussing the economic implications of change within and outside of markets, but it does not adequately communicate how uncertainty manifests itself in the estimates. This could be more explicit in the paper. The literature related to economic uncertainty seems insufficient and largely ignores uncertainty in the non-market context (i.e. ecosystem services). In addition, damages are reported relative to GDP using a baseline that appears to omit location-specific adaptation and to treat GDP similarly across locations applying the different risk profiles but without accounting for heterogeneous economic growth or adaptive capacity.

AUTHORS: We agree with the reviewer on this point, and we have improved the manuscript in this aspect. We have added a 'Limitations' section in the Methods where we discuss this and other shortcomings of the paper (some discussed also below).

Assessment of Uncertainty in Changes in Damage Relative to GDP: The analysis is excellent in accounting for potential uncertainty in the physical hazard chain related to exposure, vulnerability, and damages. The literature cited also shows the importance of adaptation and the study has extensive discussion of adaptation following the results. It seems that the manuscript can expand on the description of the uncertainty in the economic results and what drives that uncertainty. For example, Bachner et al (2022) appears to be relevant to this application. Bachner et al. (2022) seem to describe a no adaptation scenario as an implausible reference point when forecasting damage and GDP changes from sea level rise into the future. It is important to then consider how a no-adaptation dollar estimate can be interpreted since these areas will be forced to adapt out of necessity. This likely has implications for how a reader should interpret the economic results, with damage estimates representing potential upper bounds if realistic adaptation is not accounted for. Further, economic differences across locations are likely to lead to different rates of GDP growth, which could be significant over a long time horizon. That makes comparisons between GDP and expected damages less certain between locations over the long time horizon. Newell et al (2021) also provides relevant background information for explaining potential model uncertainties. Newell investigates many plausible models linking temperature and GDP. While temperature impacts are different than damages from sea level rise and erosion, that work discusses both level effects and growth effects on GDP. Newell results show estimates of GDP changes to be uncertain

because growth responses are sensitive to modeling assumptions. Since this paper compares damage estimates to GDP, this seems like a relevant topic, even if it is only referred to in the detailed methodology as a limitation and area of future research. The manuscript already mentions differences in financial capacity across locations in the discussion, so this seems like a natural extension that acknowledges the complexity of economic outcomes when faced with these risks.

AUTHORS: We would like to thank the reviewer for this constructive comment. We have added a paragraph discussing all the points raised by the reviewer. Here is the text:

‘While our analysis accounts for uncertainty in the physical hazard chain, it is important to also consider sources of uncertainty in the economic baseline used for comparison, namely GDP. Regional differences in financial capacity, institutional readiness, and long-term growth trajectories can introduce substantial variability in GDP projections, complicating cross-regional comparisons of damages relative to economic output. In addition, macroeconomic impacts of climate change are highly sensitive to assumptions about whether climate shocks affect GDP levels or long-term growth rates. Although our results focus on relative damages as a share of GDP, readers should be aware that these estimates remain sensitive to broader economic uncertainties not fully captured in the physical impact modeling. Also, our damage estimates assume of a no-adaptation scenario and thus they represent a stylized, increasingly implausible reference case intended to isolate potential physical impacts. They are meaningful as they can allow estimating the cost of inaction, but should be interpreted as a potential upper bound of damages⁵⁷.’

Assessment of Ecosystem Services

In the paper, the manuscript first provides an estimate for, and then highlights changes in, aggregate ecosystem service value from changing coastal conditions. Costanza et al. (2014) argue that for raising awareness, Total Aggregate Values applied to regional or global scales are appropriate and low precision is acceptable. I find this a reasonable argument, but the paper should be explicit in communicating this so not to mislead readers or overstate its policy applications beyond highlighting value at risk. I do have some concerns that the manuscript uses global mean values for ecosystem services rather than following some type of transfer values that are adjusted--possibly even values using a more spatially

explicit transfer function that accounts for differences in sites and populations. Johnston et al. (2021) provide details into best practices for benefit transfer. The manuscript makes no note of the assumptions necessary for the transfer of mean global values in this context or the implications of using these types of values. The manuscript really glosses over the limitations of those estimates.

AUTHORS: We fully agree with the reviewer and we discuss this point in the 'Limitations' section found in the Methods.

'In the absence of more detailed information, we assume that the value of ecosystem do not vary spatially. This limitation may be inevitable but is one of the many sources of epistemic uncertainty related to quantifying the economic benefits from ecosystems. More spatially explicit valuation methods, such as benefit transfer functions that adjust for local context, could improve accuracy in future work. Another important aspect that is omitted here is that the value of ecosystems tends to increase as they become scarcer. Another limitation of our approach is that, given the scarcity of such information, we are not considering any uncertainty in the ecosystem service value estimates. Acknowledging all the above, we emphasize that our results are intended to signal the potential magnitude of ecosystem service loss under continued coastal change.'

Additionally, the manuscript does not provide estimates of uncertainty global mean values for ecosystem services. It could still rely on mean values in analysis while also communicating variability using standard errors or confidence intervals.

AUTHORS: Same the previous comment and is discussed in the 'Limitations' section found in the Methods (see previous response for text).

Fenichel and Abbot (2014) argue that as ecosystem services become scarcer over time, the marginal value of those services can increase non-linearly. Drupp et al. (2024) extend that argument. This highlights the importance of feedbacks between ecosystems and economic behavior within and outside markets. I think it is important for readers to realize that the marginal value of ecosystem services rises with both increased scarcity and increased wealth. You could add these types of references to the detailed methodology to communicate uncertainty over long time periods.

AUTHORS: Same the previous comment and is discussed in the ‘Limitations’ section found in the Methods (see previous response for text).

Policy Relevance

This is a very important topic for these impacted regions. There will need to be significant efforts for both protection and adaptation in the study areas. While I have concerns over how policy makers should interpret the economic estimates, they provide an excellent opportunity to raise awareness. The manuscript can both clearly state the uncertainties and the extensive assets at risk.

AUTHORS: We would like to thank the reviewer once again for his positive feedback and constructive comments. We are confident that the revision is more transparent about uncertainties.

Minor Comments:

- On line 81-82, you state that these “flooded areas do not always translate into population exposure and economic impacts.” I am assuming that you are referring to market based economic impacts. This is slightly confusing since you also include ecosystem service valuations in the paper and ecosystem service values can manifest on local, regional, or global levels. This is very minor, but slight rewording may be helpful.

AUTHORS: We have rephrased the sentence:

‘so flooded areas do not always translate into population exposure and damage to infrastructure’

- Spelling: change asses to assess on line 69

AUTHORS: We have corrected the spelling error.

Overall assessment

Overall, I found this to be a very strong paper, but it is inadequate in communicating the limitations of the economic results and in providing guidance for how a reader should interpret the economic results. I recommend the authors clarify these areas both in the main portion of the manuscript and in detailed methods. With those changes, this manuscript represents a strong contribution to the climate change literature.

My Expertise

I am an environmental economist.

Citations

Bachner, G., Lincke, D., & Hinkel, J. (2022). The macroeconomic effects of adapting to high-end sea-level rise via protection and migration. *Nature Communications*, 13(1), 5705.

Costanza, R., De Groot, R., Sutton, P., Van der Ploeg, S., Anderson, S. J., Kubiszewski, I., ... & Turner, R. K. (2014). Changes in the global value of ecosystem services. *Global environmental change*, 26, 152-158.

Drupp, M. A., Hänsel, M. C., Fenichel, E. P., Freeman, M., Gollier, C., Groom, B., ... & Venmans, F. (2024). Accounting for the increasing benefits from scarce ecosystems. *Science*, 383(6687), 1062-1064.

Fenichel, E. P., & Abbott, J. K. (2014). Natural capital: from metaphor to measurement. *Journal of the Association of Environmental and Resource Economists*, 1(1/2), 1-27.

Johnston, R. J., Boyle, K. J., Loureiro, M. L., Navrud, S., & Rolfe, J. (2021). Guidance to enhance the validity and credibility of environmental benefit transfers. *Environmental and Resource Economics*, 79(3), 575-624.

Newell, R. G., Prest, B. C., & Sexton, S. E. (2021). The GDP-temperature relationship: implications for climate change damages. *Journal of Environmental Economics and Management*, 108, 102445.